# Measurement report: In-flight and ground-based measurements of nitrogen oxide emissions from latest generation jet engines and 100% sustainable aviation fuel

Theresa Harlass[1], Rebecca Dischl[1,2], Stefan Kaufmann[1], Raphael Märkl[1,2], Daniel Sauer[1], Monika Scheibe[1], Paul Stock[1], Tiziana Bräuer[1,2], Andreas Dörnbrack[1], Anke Roiger[1], Hans Schlager[1], Ulrich Schumann[1], Magdalena Pühl[1], Tobias Schripp[3], Tobias Grein[3], Linda Bondorf[3], Charles Renard[4], Maxime Gauthier[4], Mark Johnson[5], Darren Luff[5], Paul Madden[5], Peter Swann[5], Denise Ahrens[6], Reetu Sallinen[7] and Christiane Voigt[1,2]

[1] Deutsches Zentrum für Luft- und Raumfahrt, Institut für Physik der Atmosphäre, Oberpfaffenhofen, Germany
[2] Johannes Gutenberg-Universität Mainz, Institut für Physik der Atmosphäre, Mainz, Germany
[3] Deutsches Zentrum für Luft- und Raumfahrt, Institut für Verbrennungstechnik, Stuttgart, Germany
[4] Airbus Operations SAS, Toulouse, France
[5] Rolls-Royce plc. Derby, UK
[6] Rolls-Royce Deutschland, Dahlewitz, Germany
[7] Neste Corporation, Innovation, Porvoo, Finland

*Correspondence to*: Tiziana Bräuer (Tiziana.braeuer@dlr.de), Anke Roiger (Anke.Roiger@dlr.de)

**Abstract.** Nitrogen oxides, emitted from air traffic, are of concern due to their impact on climate by changing atmospheric ozone and methane levels. Using the DLR research aircraft Falcon, total reactive nitrogen ($NO_y$) in-flight measurements were carried out at high altitudes to characterize emissions in the fresh aircraft exhaust from the latest generation Rolls-Royce Trent XWB-84 engine aboard the long-range Airbus A350-941 aircraft during the ECLIF3 experiment. The impact of different engine thrust settings, monitored in terms of combustor inlet temperature, pressure, and engine fuel flow, was tested for two different fuel types: Jet A-1 and for the first time a 100% sustainable aviation fuel (SAF) under similar atmospheric conditions. In addition, a range of combustor temperatures and an additional blended SAF were tested during ground-based emission measurements. For the data measured during ECLIF3, we confirm that the $NO_x$ emission index increases with increasing combustion temperature, pressure and fuel flow. We find that as expected, the fuel type has no measurable effect on the $NO_x$ emission index. These measurements are used to compare to cruise $NO_x$ emission index estimates from three engine emission prediction methods. Our measurements thus help to understand the ground to cruise correlation of current engine emission prediction methods while serving as input for climate modelling, and extending the extremely sparse data set on in-flight aircraft nitrogen oxide emissions to newer engine generations.

## 1 Introduction

Aviation is a steadily growing transport sector (Lee et al., 2021), even despite the short-term drop during the COVID-19 pandemic (Le Quéré et al., 2020; Schumann et al., 2021; Voigt et al., 2022). As a result, emissions from air traffic are also expected to increase continuously and higher aircraft and engine efficiencies are surpassed by the overall air traffic growth. The exhaust of an aircraft engine burning conventional kerosene is constituted on average of ~3.16 kg carbon dioxide ($CO_2$) and ~1.23 kg water vapour ($H_2O$) per kg of fuel burned (Lee et al., 2021). Further emissions depend on the engine type, power settings, combustor technology and fuel composition and include nitrogen oxides ($NO_x$) as the sum of nitric oxide (NO) and nitrogen dioxide ($NO_2$), carbon monoxide (CO), unburnt hydrocarbons ($C_xH_y$), sulphur dioxide ($SO_2$) and soot (or non-volatile particulate matter, nvPM).

Most recent estimates (Lee et al., 2021) state that aviation since its historical beginnings contributes with +100.9 (55-145) mW $m^{-2}$ to about 4 to 5% to total anthropogenic effective radiative forcing (ERF) (Grewe et al., 2019; Lee et al., 2021). Thus, air traffic emissions have a warming effect on climate. Contrail cirrus hereby represent the largest share of aviation ERF with ~57%, followed by $CO_2$ (~34%) and $NO_x$ emissions (~17%; (Lee et al., 2021)). $NO_x$ emissions from aviation contribute indirectly to anthropogenic ERF via a short-term warming and a long-term cooling effect (Brasseur et al., 1996; IPCC, 1999; Köhler et al., 2008; Lee et al., 2010; Dahlmann et al., 2011; Grewe et al., 2019; IPCC, 2021; Lee et al., 2021; Skowron et al., 2021; Terrenoire et al., 2022). The net ERF from $NO_x$, in sum, is however positive with 17.5 (0.6-29) mW $m^{-2}$ (Lee et al., 2021). Other studies indicate that the contribution of $NO_x$ in aviation ERF may be even higher due to simplifications in the methodology of previous estimates (Grewe et al., 2019). In the future, aviation $NO_x$ emissions may have a net negative ERF, as the effect of emission of aircraft $NO_x$ depends strongly on background emissions (Skowron et al., 2021).

Up to now, there is a lack of experimental measurements of $NO_x$ emissions at cruise altitudes from state-of-the-art jet engines. Thus, in the joint Emission and CLimate Impact of alternative Fuels 3 (ECLIF3) project, direct ($NO_x$, CO, nvPM, $H_2O$) and indirect aircraft emissions (ice particles, vPM) were measured at high altitudes to understand and assess the impact of modern air traffic on the atmosphere, see also Märkl et al. (2023) and previous related projects as in Voigt et al. (2021). In this study, we present a comprehensive set of $NO_y$ emission measurements performed in the exhaust of the Airbus A350-941 with Rolls-Royce Trent XWB-84 engines. We derive emission indices from in-flight measurements and compare them to ground-based measurements and three different engine emission prediction methods. We further investigate the effect of the Airbus aircraft burning 100% sustainable aviation fuel (SAF). Replacing conventional kerosene with SAF is one promising approach to reduce engine soot emissions, ice crystal number concentrations in contrails and the related climate impact, all in addition to a potential reduction of the $CO_2$ footprint in the life cycle analysis (Voigt et al., 2011; Moore et al., 2017; Kleine et al., 2018; Bräuer et al., 2021b; Bräuer et al., 2021a; Voigt et al., 2021).

## 2 Materials and Methods

### 2.1 In-flight $NO_y$ and $CO_2$ measurement methods during ECLIF3

The DLR-operated research aircraft Falcon (reg. D-CMET, Dassault Falcon 20-E5), a twin-engine jet, was used as the airborne measurement platform. The in-flight instrumentation consisted of several cabin-mounted trace gas ($NO_y$, CO, $CO_2$, $H_2O$) and aerosol instruments with their sample inlets located on the upper fuselage, see Figure 1b. Also, cloud particle probes were mounted in underwing pods to measure ice particle number concentrations and size distributions in ambient conditions (Märkl et al., 2023). Here, we focus on the $NO_y$ and $CO_2$ measurement instruments as they are needed to derive the respective $NO_x$ emission indices ($EI(NO_x)$).

A well-established technique for measuring reactive nitrogen species, as employed in the present paper, includes their catalytic conversion to NO and subsequent detection using chemiluminescence technique (ICAO, 2017). In general, chemiluminescence detectors (CLD) are widely used in atmospheric monitoring because they are sensitive with a detection limit as low as a few parts per trillion (ppt). In CLD, a sample of air passes through a reactor where NO is excited to $NO_2^*$ by the reaction with high concentrations of $O_3$ produced by an ozone generator (Ridley and Howlett, 1974; Drummond et al., 1985). When the excited $NO_2^*$ molecules return to their ground state, the light emitted by the chemiluminescence reaction is proportional to the concentration of NO in the sample. By using selective converters directly upstream of the measurement chamber different reactive nitrogen species are converted to NO and subsequently detected by the CLD (Bollinger et al., 1983; Fahey et al., 1985). Within DLR, different types of CLD detectors have been used for atmospheric background measurements (Schlager et al., 1997; Ziereis et al., 2000; Schmitt, 2003; Stratmann et al., 2016; Ziereis et al., 2022) as well as in-plume detections (Schulte and Schlager, 1996; Schlager et al., 1997; Roiger et al., 2015). Aboard the Falcon a heated gold converter (T = 290 °C) with hydrogen ($H_2$) as reducing agent catalytically reduces all $NO_y$ compounds to NO. $NO_y$ is the sum of $NO_x$ (NO and $NO_2$) and all other reactive nitrogen species as e.g. nitric acid, peroxyacetylnitrate (PAN) etc. The instrument offers no measurement of NOx or the NOx/NOy ratio. The time resolution of the instrument is ~1 Hz with a detection limit of ~0.5 ppb. Under normal operating conditions the detector runs into saturation above mixing ratios of ~1000 ppb or ~1e6 counts per seconds. In order to measure at higher concentrations as expected in the exhaust plumes, a dilution system was integrated supplying zero air to the sample air in a ratio of 1/1 prior to the measurement. The $NO_y$ measurement accuracy ranges between ~5 ppb near the detection limit and ~490 ppb at highest detected mixing ratios of ~4 ppm. It includes the following uncertainty parameters typical for CLD instruments, for details see (Stratmann, 2013): the sensitivity of the instrument (841 ± 95 counts/ppb), the efficiency of the $NO_y$ converter (98.7 ± 1.5% at low and ± 30% at high concentrations), the instrumental interferences due to desorption processes and dark current (36 ± 118 counts), the statistical uncertainty of count rates (150-1600 counts), the uncertainty in the calibration standard (1% as stated by the manufacturer), the uncertainty due to the dilution system (7-21%) and due to a second instrument stage above 1e6 counts per seconds (i.e. 300 ppb).

A high frequency (~10 Hz) non-dispersive infrared gas analyser (LI-7000, LI-COR Biosciences) was used aboard the Falcon for in-plume $CO_2$ detections to be able to capture the small-scale variability of the plume. The LI-7000, uniquely modified in-

house for aircraft deployment, uses two measurement chambers to detect $CO_2$: one is constantly flushed with dry zero air, the other is supplied with ambient air. The difference in absorption of infrared radiation passing through both cells is used to determine the absolute absorption and the absolute $CO_2$ mixing ratio (LI-COR, 2007). In the post processing, the $CO_2$ mixing ratio is corrected for dilution effects (LI-COR, 2003) and reported as dry air mole fraction. The $CO_2$ accuracy of the LI-7000 is independent with respect to the measured mixing ratios and is around 0.2 ppm. This includes the reproducibility of the calibration standards (0.08 ppm), the precision (0.08 ppm) and the uncertainty of water vapour measurement and thus the dilution correction (0.16 ppm). An occurring trend of the instrument response with instrument temperature and time is compensated by frequent zero measurements every 30 minutes during the flight. Absolute $CO_2$ mixing ratios and background values were cross checked with a second instrument aboard the Falcon, a cavity ring-down spectrometer (CRDS, Picarro G2401-m), due to its stable instrument performance (Fiehn et al., 2020; Klausner et al., 2020; Klausner, 2020).

## 2.2 In-flight emission sampling strategy

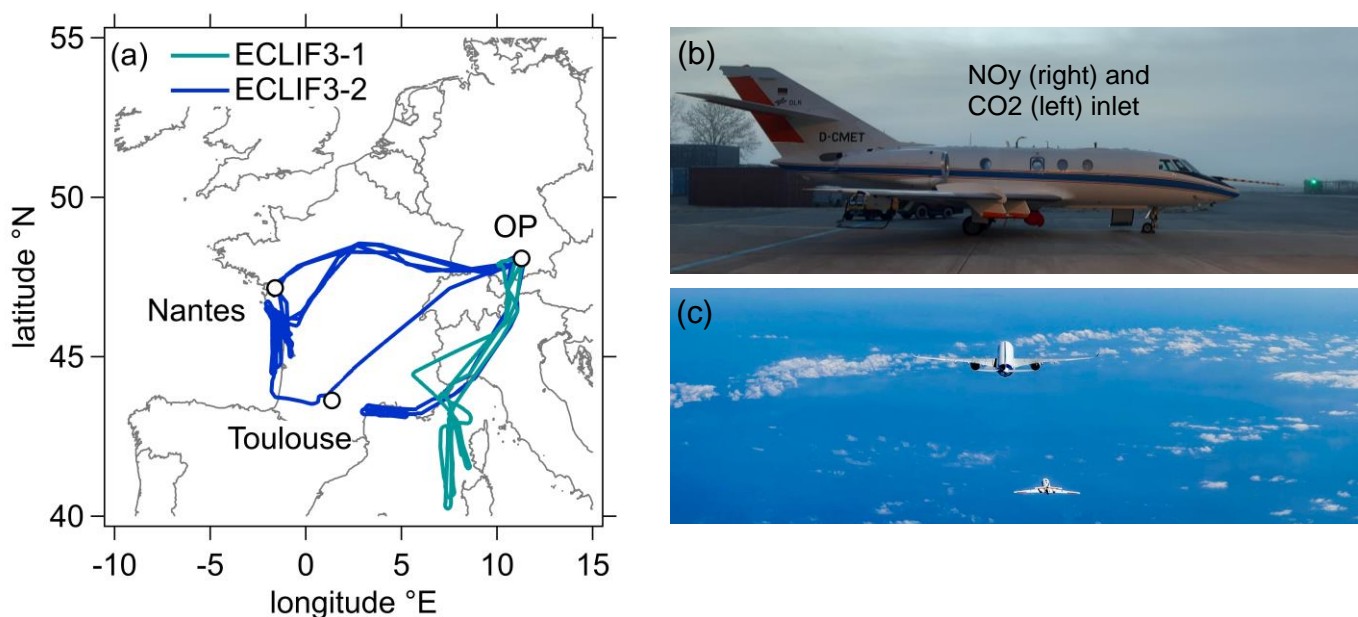

**Figure 1: (a) Map of the Falcon flight routes, (b) inlet positions for trace gas sampling aboard the Falcon and (c) Falcon aircraft catching up to the Airbus A350 to perform emission measurements, © Airbus 2021, photo by S. Ramadier.**

In-flight measurements at high altitudes were carried out under the framework of ECLIF3 in April 2021 (referred to as ECLIF3-1) and in November of 2021 (referred to as ECLIF3-2). This joint project was led by DLR and Airbus with Rolls-Royce, Neste, the University of Manchester and the National Research Council Canada (NRC) as partners. The objective of these flight experiments was to characterize emissions behind an Airbus A350-941 (reg. F-WXWB) with Rolls-Royce Trent XWB-84 engines (ECLIF3-1: engine number 21004, ECLIF3-2: engine number 21012). This twin-engine aircraft is used for long-range distances by many operators since it is more fuel efficient than a four-engine long-range aircraft and older two-engine aircraft.

During the mission flights, the Airbus aircraft was able to feed the engines from separate fuel tanks. Hence, measurements
with two different fuels could be performed within single measurement flights which allowed probing of emissions at similar
atmospheric conditions. As reference fuel, conventional Jet A-1 was used and provided by the local fuel supplier TotalEnergies.
As sustainable aviation fuel, a 100% HEFA-SPK (Hydroprocessed Esters and Fatty Acids Synthetic Paraffinic Kerosene) made
from sustainably sourced renewable waste and residues such as used cooking oil and other waste fat was provided by the
project partner Neste.

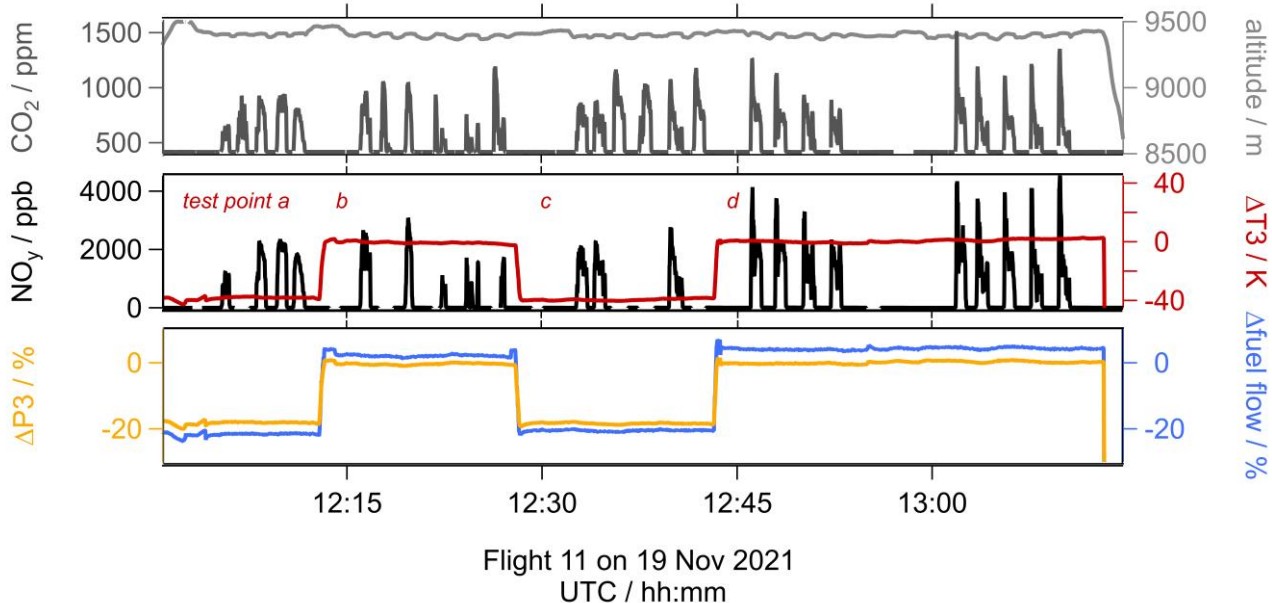

**Figure 2: Emission measurement sampling sequence during the flight on 19 Nov 2021. Trace gases ($CO_2$, $NO_y$) and flight altitude as measured aboard the Falcon. Change in Airbus aircraft engine parameters (delta fuel flow, delta T3 temperature, delta P3 pressure) indicate different engine test points.**

Emission measurements in the near-field were performed as close as possible to the Airbus aircraft at distances between 65 to
470 m (mean of 260 m), resulting in young plume ages of 0.3 to 3.5 s (mean of 0.8 s) in order to sample the fresh emissions
before they undergo chemical processing in the atmosphere. Furthermore, it was aimed to study the emissions for various fuels
at different combustor inlet parameters (temperature T3, pressure P3, engine fuel flow / fuel-to-air ratio) by varying the engine
power settings and flight altitudes in defined but variable test point sequences. In total, six emission chase flights were
performed along the western and southern French coast in temporary reserved air space, as shown in Figure 1a. The Falcon
was mainly based in Oberpfaffenhofen (OP), Germany, and for a 5-day period in Toulouse, France, to minimize transfer flight
times to the Airbus test area. With a take-off from OP, the Falcon had to be refueled in Nantes prior to the measurement flights
along the Atlantic coast.
An example of an emission time series is shown in Figure 2. Measurements took place in the near-field with the Falcon aircraft
flying behind the Airbus aircraft under non- or only short-lived contrail-formation conditions. Usually, emissions were sampled

from the right-hand side engine which was operated at different, well-defined cruise combustor inlet temperature conditions. The Falcon is the slower flying aircraft with a maximum cruise speed of ~200 m s$^{-1}$. To ensure that both aircraft speeds match in order to maintain the close distance, the Airbus aircraft had to adjust its speed by operating the left engine at a lower thrust.

Hence, the test conditions probed in this study are not fully representative of typical cruise conditions. In addition, the Airbus aircraft also typically flies at higher altitudes (above FL350) compared to the altitudes which were sampled within this project (FL180 to 360). Generally speaking, this means that, at fixed T3 and DISA (Delta International Standard Atmosphere), the EI(NO$_x$) measured at the lower test altitudes tend to be higher than the actual EI(NO$_x$) at typical cruise altitudes (due to higher P3 at lower altitudes). Nevertheless, we assume that above FL300, this effect is small and the measurements can be directly

compared to predictions at test conditions.

To ensure that the ceiling with the trace gas inlets is directly located in the exhaust plumes, the Falcon approached the exhaust trail from slightly lower altitudes. This position was held for ~50 s, then the Falcon dropped down to measure atmospheric background conditions for ~30 s. This alternating sequence was repeated 3-5 times for each test point with fixed engine parameters. Large enhancements (Δ) of CO$_2$ and NO$_y$ were clearly visible with values typically between 100-800 ppm (ΔCO$_2$)

and 500-4000 ppb (ΔNO$_y$) above mean atmospheric background values of 414-419 ppm and 0.6-4 ppb, respectively.

In Section 3.3, we will use additional in-flight far-field measurements from ECLIF3 for a comparison with engine emission prediction methods to achieve more representative cruise conditions. Far-field measurements hereby represent measurements at distances between 6 to 38 km (mean of 20 km) in the aged exhaust (30 to 390 s). Representative cruise conditions aim at similar Mach numbers which can only be achieved when the Airbus is flying at its typical speed.

**2.3 Ground-based NO$_y$ and CO$_2$ measurement methods during ECLIF3-2**

In-flight measurements during ECLIF3-2 are complemented by ground-based measurements of the same engine performed in Toulouse, France, in October 2021. The CO$_2$ concentration was monitored using two non-dispersive infrared (NDIR) sensors (LI-7200RS and LI-850 by LI-COR Biosciences). The NO$_y$ concentration was monitored using two chemiluminescence detectors (CLD64 and CLD700 by ECO PHYSICS), which use converters operated at 400°C. The instruments were calibrated

with certified calibration gases before each ground-based test run. Figure 3 shows the measurement setup for the ground-based measurements. Sampling was performed with a stainless-steel probe inlet at a height of 2.6 m and a distance of 24.5 m to the right-hand side engine exit plane. The exhaust was transported from the probe inlet via a 40 m heated stainless-steel line to a manifold that allowed even distribution of air to the different measuring instruments.

During the ground-based measurements, three fuel types were tested (see Table 1). The measurements with fossil Jet A-1 were

165 performed on 22 October 2021 (ambient temperature: 12.2°C - 14.4°C; ambient relative humidity: 87% - 81% during test run). The neat HEFA-SPK and an additional fuel blend (HEFA-SPK blended with a different Jet A-1) were tested on 23 October 2021 (ambient temperature: 9.1°C - 15.2°C; ambient relative humidity: 47% - 96% during test run). Four test points with Jet A-1 were measured on both days to identify any biases from changes in ambient conditions or different probe alignment.

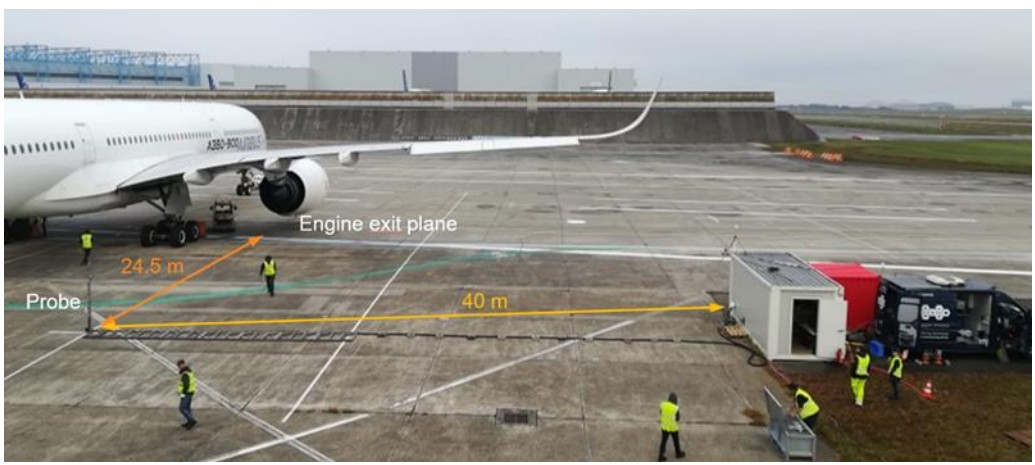

**Figure 3: Setup of ECLIF3-2 ground-based measurements behind the A350 with Stainless Steel Probe and Heated Stainless Steel Line. White Container: particle measurement instruments for analysing particulate matter in the exhaust. Red Container: location of the Proton Transfer Reaction Mass Spectrometer (PTRMS) for the detection and quantification of volatile organic compounds (VOCs). Blue Mobile Laboratory: measurement devices for gaseous species, including instruments for measuring $NO_y$ and $CO_2$.**

The test grid of the ground-based measurements consisted of varied T3 temperatures for the combustor stage settings, ranging from idle to maximum power. Each test point was stabilised for a few minutes and after reaching a stable T3, sampling was performed for 5-6 minutes. In case of the highest thrust settings, the sampling time was reduced to 2-3 minutes. The sampling height and distance were chosen to ensure representative capture of the exhaust plume, while the heated line helped prevent condensation and potential losses or alterations in the sample composition. The use of various measuring instruments and

redundant systems further enhanced the accuracy and reliability of the data collected. The different gains of the chemiluminescence detectors allowed for precise measurement of a wide range of $NO_y$ concentrations, while the multiple measuring points of the NDIR systems ensured continuous monitoring of $CO_2$ concentrations. The selection of T3 temperatures and the varying operational conditions of the combustor stages from idle to maximum power allowed for a comprehensive analysis of emissions under different operating conditions.

**2.4 Emission index calculation and plume definition**

In order to quantify exhaust emissions from aircraft, the most common metric is the so-called emission index (EI), i.e. an emission quantity per mass of burned fuel. The $NO_x$ emissions index ($EI(NO_x)$) is defined by convention in mass units of $NO_2$ ( Voigt et al., 2012; ICAO, 2017; ICAO, 2023), hence the sum of NO and $NO_2$ in ambient air is calculated as if all NO was in the form of $NO_2$. Several studies discuss the composition of the different nitrogen species in the engine exhaust e.g. (Kärcher

et al., 1996; Tremmel et al., 1998; Bradshaw et al., 2000; Wormhoudt et al., 2007; Voigt et al., 2012) agree that at high engine power settings $NO_x$ in the exhaust is dominated by NO (>80%). With growing plume age, the NO and $NO_2$ ratio is determined by an equilibrium of the reaction of NO and $O_3$, forming $NO_2$ and the photolysis of $NO_2$ (Tremmel et al., 1998). In addition,

small amounts of $HNO_2$, $HNO_3$ and HONO are formed in the ageing plume from the $NO_x$ emissions (Jurkat et al., 2011, Lee at al., 2011). During ECLIF3, only $NO_y$ and no $NO_x$ concentrations were measured aboard the Falcon. $NO_x$ concentrations are expected to be close to $NO_y$ and the fraction of nitrogen acids in the exhaust gas is assumed to be smaller than the $NO_y$ mean measurement accuracy. Hence, all reactive nitrogen species in the exhaust are detected and related to the initial $NO_x$ emissions. For each individual plume encounter $EI(NO_x)$ is derived based on the inert dilution tracer $CO_2$ via Eq. (1) following Schulte et al. (1997):

$$\text{EI}(NO_x) = \frac{\int \Delta NOy}{\int \Delta CO_2} * EI(CO_2) * \frac{M(NO_2)}{M(CO_2)}, \tag{1}$$

where $\int \Delta NOy$ and $\int \Delta CO_2$ is the integrated enhancement above an atmospheric background level; $EI(CO_2)$ is the emission index of $CO_2$ dependent on the fuel used (see Table 1); and $M(NO_2)$ and $M(CO_2)$ are the molar masses of $NO_2$ (46.0055 g mol$^{-1}$) and $CO_2$ (44.0095 g mol$^{-1}$).

An individual in-flight plume encounter denotes a plume crossing in a time series where enhancements of $NO_y$ and $CO_2$ start exceeding background level variations, denoting the plume beginning, and the subsequent return to atmospheric background level, denoting the plume end. Between plume beginning and end a minimum 7 second plume length threshold was chosen to exclude accidental plume encounters. Further, plumes were rejected due to Airbus aircraft engine instability (e.g. variability of T3 greater than $\pm 2$ K) and/or low correlation between $CO_2$ and $NO_y$ measurements (R<0.7). The individual plume crossings of the background-corrected mixing ratios of each species are integrated over time (and hence, over the horizontal extent of the plume) to account for the inlet positions (about 30 cm difference in the horizontal direction), tubing lengths and different instrument response times. Due to the variable position of the Falcon aircraft within the turbulent exhaust plume and thus variable plume dilution values, it is necessary to normalize $NO_y$ mixing ratios to the measured $CO_2$ concentration. $CO_2$ hereby acts as a chemically inert species to determine the dilution of the engine emissions at the measurement point. $CO_2$ emission indices are a fuel-dependent metric and are derived from the hydrogen and carbon content of the fuel under the assumption that it is completely burnt and all available carbon is converted to $CO_2$. The emission index $EI(CO_2)$ can then be calculated via Eq. (2) following Moore et al. (2017):

$$\text{EI}(CO_2) = \frac{RT}{pV_m} * \frac{M(CO_2)}{M(C) + \alpha M(H)}, \tag{2}$$

where $R$ is the ideal gas constant (8.31 J mol$^{-1}$ K$^{-1}$); $T$ and $p$ are the temperature (273.15 K) and pressure (101325 Pa) at standard conditions; $V_m$ is the molar volume at standard conditions (0.0224 m$^3$ mol$^{-1}$); $M(C)$ and $M(H)$ are the molar masses of carbon (12.01 g mol$^{-1}$) and hydrogen (1.01 g mol$^{-1}$); and $\alpha$ is the hydrogen-to-carbon molar ratio of the fuel (as calculated based on Table 1). As the batches of the fuel supplier varied between the two measurement experiments, also the fuel properties slightly varied. Their characteristic hydrogen and carbon contents are listed in Table 1 together with the calculated emission index of $CO_2$. The aromatics content was partly determined by GCxGC (mass based) measurements due to the contents being below the ASTM D6379 (vol based) detection limits. Samples were taken at different points of the fueling process. The hydrogen content was measured via low resolution nuclear magnetic resonance spectrometry (ASTM D3701 with a repeatability of 0.09% and reproducibility of 0.11%). The carbon content is assumed to add up to 100 mass% with the hydrogen

content and sulphur content (not listed), and was cross checked via ASTM D5291 (which has a repeatability of 0.94% and reproducibility of 2.42%). The energy content of the fuels does not differ significantly.

Table 1: Fuel properties during the measurement experiments. EI($CO_2$) is estimated following Moore et al. (2017). Last column denotes the test points used for EI($NO_x$) calculation.

| | Method | Fuel type | Density at 30°C / g cm$^{-3}$ | Aromatics content / vol% | Hydrogen content / mass% | Carbon content / mass% | EI($CO_2$) / g kg$^{-1}$ | Test Points* |
|---|---|---|---|---|---|---|---|---|
| **ECLIF3-1** | In-flight | Jet A-1 | 0.7800 | 13.4 | 14.08 | 85.90 | 3149 | 8 |
| | In-flight | HEFA-SPK | 0.7618 | 0.41** | 15.11 | 84.89 | 3111 | 11 |
| **ECLIF3-2** | In-flight & ground-based | Jet A-1 | 0.7767 | 13.4 | 14.25 | 85.74 | 3142 | 25 & 20 |
| | In-flight & ground-based | HEFA-SPK | 0.7608 | 0.02** | 15.18 | 84.82 | 3108 | 13 & 13 |
| | Ground-based | Blend | 0.7781 | 10.8 | 14.39 | 85.56 | 3137 | 12 |

* For in-flight measurements test points are defined as plume encounters, for ground-based measurements test points are defined as an averaged measurement sequence at stable T3 operating conditions.
** Aromatics content determined by GCxGC analysis in mass%.

## 2.5 Ground-based ICAO Aircraft Engine Emissions Databank

The reporting of emissions in the vicinity of airports is mandatory for engine manufacturers during certification processes for individual engine types and reports are voluntarily made publicly available at the International Civil Aviation Organization (ICAO) Aircraft Engine Emissions Databank. For that, a landing and takeoff (LTO) cycle is defined to derive emissions during taxi-out/taxi-in, approach, climb and takeoff in a consistent manner (ICAO, 2017). While varying the engine power settings, and thus the fuel flow rate at a test stand, emissions at sea level conditions are measured. Thrust levels of 7%, 30%, 85% and 100% hereby correspond to taxi-in/taxi-out, approach, climb and takeoff conditions.

As $NO_x$ emissions highly depend on temperature and pressure in the engine combustor, they are derived for different thrust settings related to different flight phases at certification ground test for International Standard Atmosphere (ISA) conditions. In general, $NO_x$ emissions increase with increasing power and increasing fuel flow, see Figure 4. Thermal NO formation, first described by Zeldovich in 1946 and therefore also referred to as Zeldovich mechanism, is one of the main sources of nitrogen oxides in combustion dominating at high pressure and temperature conditions typical for a jet engine (Lavoie et al., 1970). However, as chemical equilibrium of the thermal NO formation route is not reached within typical timescales of a combustor, total formed $NO_x$ in a modern rich burn combustor is strongly dependent on the quick quenching and mixing of the hot streams of the primary combustion zone. Combustor design optimised for $NO_x$ emissions aims for enhanced mixing and reducing

residence time in areas of high temperature. For a given combustor design, $NO_x$ emissions depend on operating conditions as pressure, temperature and local air-to-fuel ratio in the combustor. $NO_x$ in a gas turbine combustor, i.e. at high pressure and temperature, is predominantly formed via thermal NO pathway. Thermal $NO_x$ formation rate increases exponentially with temperature and further depends on pressure (Gokulakrishnan and Klassen, 2013). In comparison to all ~560 engine types covered by the ICAO Aircraft Engine Emissions Databank (depicted in grayish/black; (EEDB, 2021)), $NO_x$ emissions from

the Rolls-Royce Trent XWB-84 engine (depicted in blue) are rather at the upper limit when considering the thrust settings, but typical for a modern large engine powering the long range planes. This engine is specifically designed for a modern long-range aircraft being the most fuel-efficient large aero-engine in revenue service, and therefore it operates at high pressure ratios and combustor exit temperatures to deliver the required thrust at high fuel efficiency. Therefore, $NO_x$ emissions tend to be higher than for engines operating at lower overall pressure ratio (OPR).

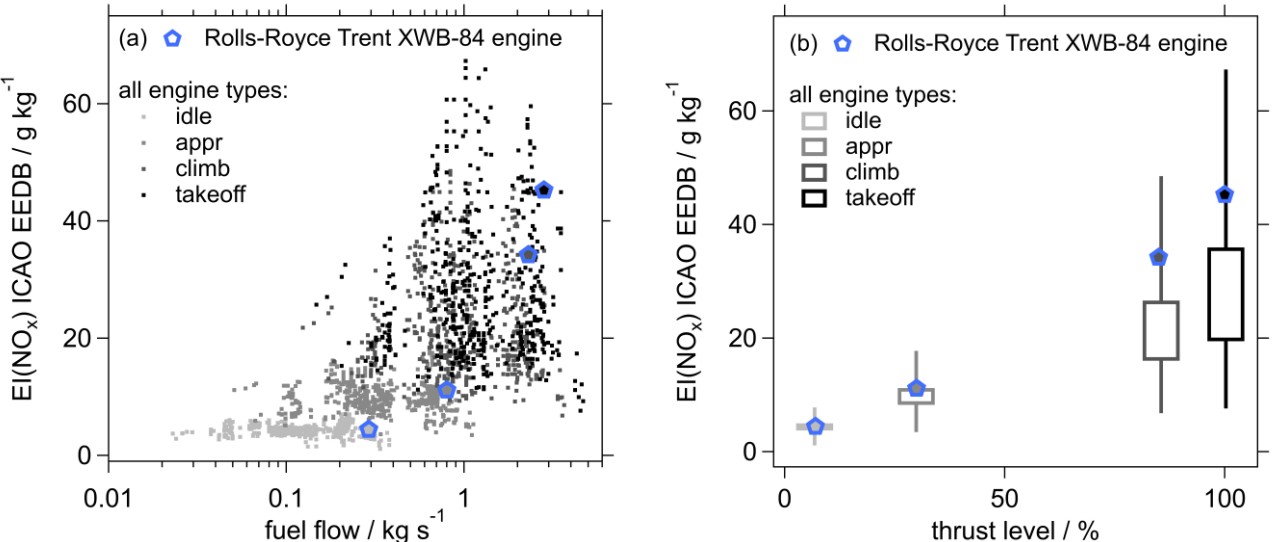

**Figure 4: EI($NO_x$) based on the entire ICAO Aircraft Engine Emissions Databank v28B (EEDB, 2021). Panel (a) shows the dependency of EI($NO_x$) on fuel flow, panel (b) on thrust. The values listed for the Rolls-Royce Trent XWB-84 engine, which was the focus during ECLIF3, are highlighted in blue.**

The certification engine emission data cover a standardised LTO cycle intended to cover local air quality. To predict EIs of

260 the whole flight envelope, different modelling approaches exist, which use ground test data and correct them to atmospheric conditions at the respective flight level, flight phase and engine thrust setting, as e.g. done when applying the Rolls-Royce model or Fuel Flow correlation methods as described in the next section.

## 2.6 Engine emission prediction methods: P3T3 and Fuel Flow methods

Based on the set of emissions data determined during the engine certification process, one can derive emission indices over

265 various flight profiles using engine performance and engine emission prediction methods. A modern high-bypass turbofan engine in general follows a Brayton or Joule Cycle, where, after a fan, two air streams are separated: in the core of the engine,

pressurized air and fuel are burned (temperature is increased by the heat release in the combustion chamber), and passed through a turbine, and together with the larger bypass air, propulsion thrust is produced. The subsequent positions are characterized and defined by total temperature and pressure regions: T2/P2 at the fan inlet, T24/P24 at the low-pressure compressor outlet, T3/P3 at the high-pressure compressor outlet and T5/P5 at the low-pressure turbine outlet. However, input data such as e.g. P3 and T3 are proprietary and thus not publicly available.

Engine performance data or models can be used to directly predict emissions at altitude, which is called P3T3 method (DuBois and Paynter, 2006). Manufacturers thus have developed their own prediction methods for non-LTO conditions based on correlations derived from empirical rig data to correct for the effect of a change in combustor inlet pressure (P3) and combustor Air Fuel Ratio (AFR) at a fixed combustor inlet temperature (T3). These correlations are typically referred to as P3T3 methods as often the AFR exponent may be set to zero. However, as $NO_x$ sensitivity to AFR depends on the stochiometric distribution within the combustion zones of a rich burn combustor, it may also be set to some small number to reflect more $EI(NO_x)$ at richer AFRs (towards higher power), as done for this comparison where AFR is set to 0.5. The validation of these methods has proven to be accurate within 10%.

The $EI(NO_x)$ must further be corrected to the lower pressure at altitude compared to sea-level static $(P3_{sealevel}/P3_{altitude})^y$. The exponent y is unique for every engine and derived by the manufacturers, but commonly ranges between 0.2 to 0.5 with typically 0.5 for a rich burn combustor being used. In addition, $NO_x$ emissions are also dependent on the ambient humidity as the additional heat capacity of the water reduces combustion temperature, and thus the $NO_x$ formation rate. Either actual humidity measurements are needed or a reference humidity of 60% is assumed (ISO 5878). In-flight measurements of relative humidity in the ECLIF3 test areas show values between 30 and 70%, hence the reference value is a reasonable assumption. However, this relative humidity at cold ambient temperature at altitude relates to a much lower absolute humidity by mass compared to ground reference condition of 6.34 g/kg. Based on the certification humidity correction formula for $EI(NO_x)$, cruise predictions must include a humidity correction of around +12%.

The direct P3T3 method requires proprietary engine data, which are not available for modelers. Simplified methods were developed to estimate $NO_x$ emissions relating in-flight fuel flow at altitude to publicly available fuel flow data and corresponding $EI(NO_x)$ in the ICAO Aircraft Engine Emissions Databank. Such fuel flow methods (FFM) provide corrections to the different flight conditions such as altitude, humidity and Mach number (Deidewig et al., 1996; Döpelheuer and Lecht, 1999; DuBois and Paynter, 2006). The ground-based values, as reported by the ICAO Aircraft Engine Emissions Databank, are logarithmically fitted with respect to fuel flow. For $EI(NO_x)$, a power function fit leading to linear relations in a log-log plot is used between two points. Previously published comparisons with in-flight $NO_x$ measurements of older engines have shown that predictions based on fuel flow models and in-flight measurements agreed on average within ±12% (Schulte et al., 1997). For conventional rich burn combustors fuel flow methods like Boeing FFM2 (DuBois and Paynter, 2006) are able to predict altitude emissions within 10% compared to the full proprietary P3T3 method (Norman, 2003; SAE, 2009).

In this study, we use the calculated $EI(NO_x)$ of three different engine emission prediction methods and compare them to the in-flight measurements: (a) the Boeing Fuel Flow Method2 (BFFM2) with which Rolls-Royce estimated $EI(NO_x)$ at the tested

operating conditions; (b) a method based on the Fuel Flow Method2 (DuBois and Paynter, 2006) adapted and improved by DLR (aptFFM2) as described in Teoh et al. (2022); and (c) the Rolls-Royce in-house P3T3 method (P3T3). For the fuel flow, actual measurement data and the EI(CO2), according to Table 1, served as input. Differences between aptFFM2 and BFFM2 method are that the aptFFM2 method uses total pressure and temperature, therefore calculating the effect of aircraft speed, while the BFFM2 method uses ambient pressure and Mach number (Schaefer and Bartosch, 2013).

\* as coded in December 2022 in python by Roger Teoh and Marc Stettler at Imperial College and converted in 2022 to Fortran by our team at DLR.

## 3 Results and Discussion

In-flight measurements aboard the DLR research aircraft Falcon were carried out in the framework of the ECLIF3 project in 2021. Based on these in-flight measurements at high altitudes we quantify exhaust emissions by inferring emission indices of $NO_x$ valid for the young exhaust of a long-range Airbus A350-941 aircraft with latest generation Rolls-Royce Trent XWB-84 engines. For the first time, the Airbus aircraft was fuelled with 100% HEFA-SPK (Airbus, 2021b, a; Rolls-Royce, 2021). The aircraft was able to switch between Jet A-1 and HEFA-SPK during the flight mission, hence, the impact of fuel effects could be measured in comparable atmospheric conditions. Furthermore, different engine power settings, e.g. compressor exit temperatures and pressure, as well as engine fuel flows / fuel-to-air ratio, were studied at high altitudes. Due to the confidential nature of detailed aircraft and engine parameters, we do not relate the derived emission indices to absolute values of engine sensitive parameters, but use a delta notation ($\Delta$) to a representative observed value of the flight mission.

### 3.1 In-flight ECLIF3 $NO_x$ emission indices

The emitted $NO_x$ of an aircraft engine is dependent on actual thrust and resulting combustor conditions as inlet temperature T3 and pressure P3, as well as local air-to-fuel ratio. Hence, we present EI($NO_x$) values depending on different source engine parameters, as listed in Table 2.

**Table 2: Normalisation of source engine parameters.**

| Abbreviation | Parameter | Unit | State within individual test points |
|:---:|:---:|:---:|:---:|
| $\Delta T3$ | total temperature at HPC exit relative change to a representative mission value | K | controlled adjustment |
| $\Delta P3$ | total pressure at HPC exit relative change to a representative mission value | % | changes with T3 adaptations |
| $\Delta FF$ | fuel flow rate per engine relative change to a representative mission value | % | changes with T3 adaptations |

| Mach | Mach number ratio of true air speed (TAS) and local speed of sound | - | constant |
| T2 | total temperature at fan intake | % | constant |

Since the temperature at the exit of the high-pressure compressor (HPC), T3, is one of the major parameters affecting
emissions, engine throttle was set in order to achieve various levels of T3 in the cruise range (low, mid and high-power cruise).
This led to a different P3 and fuel flow levels, while Mach number and T2 were held roughly constant during different test
points by adjusting the second engine.

The dots in Figure 5a show emission indices derived from near-field measurements on 19[th] of November 2021 at flight level
(FL) 310 for varying ΔT3. Two different T3 settings and fuel types (Jet A-1 and 100% HEFA-SPK) were probed during that
flight. The flight altitude (9465 ± 10 m), Mach number (0.62 ± 0.003) and T2 (± 1%) were held constant within the different
test points, while T3 was increased by ~40 K. This led to a simultaneous increase in P3 by ~19% and fuel flow by ~26%. First,
at ΔT3 of approx. -40 K the mean $EI(NO_x)$ from Jet A-1 (16.2 ± 0.3 g kg-1) and HEFA-SPK (15.6 ± 0.2 g kg-1) agree within
their error estimates. Hence, no statistically significant impact of fuel type on $EI(NO_x)$ can be detected, although the increased
hydrogen content of HEFA-SPK could have an impact on the flame temperature and therefore on the NOx emissions (Gleason
and Martonet, 1980; Lefebvre, 1984; Yelugoti and Wang, 2023; Alabaş and Çeper, 2024)Second, the mean $EI(NO_x)$ for Jet A-
1 increases over the T3 range by ~20% (to 19.1 ± 0.4 g kg-1), the mean $EI(NO_x)$ for HEFA-SPK by ~17% (to 18.3 ± 1.7 g kg-
1). Therefore, it can also be concluded that both fuels show similar sensitivities to combustion conditions as temperature and
pressure as expected.

The squares in Figure 5a represent measurements on 14[th] and 16[th] of April 2021 at FL360 to underline the two measurements
above with additional measurement points on a different day and at different flight altitude. At the ΔT3 setting of approx. -15
K the mean $EI(NO_x)$ for Jet A-1 and HEFA-SPK again agree within uncertainties. HEFA-SPK, in addition, was probed at two
higher T3 settings (approx. 0 K and approx. 15 K) and shows also an increasing trend in mean $EI(NO_x)$ from 16.7 ± 0.5 to 18.4
± 1 and 22.0 g kg-1. The spread of individual $EI(NO_x)$ in Figure 5a may be due to sensitivity of $NO_x$ to ambient conditions.
The standard deviations from the means in $EI(NO_x)$ for FL310 are between 3-8% for Jet A-1 and around 2-11% for HEFA-
SPK. The standard deviations from the means for $EI(NO_x)$ for FL360 are up to 15 % for Jet A-1 and 3-6 % for HEFA-SPK.
However, these internal variabilities are still smaller than the increasing trend of the mean values with T3.

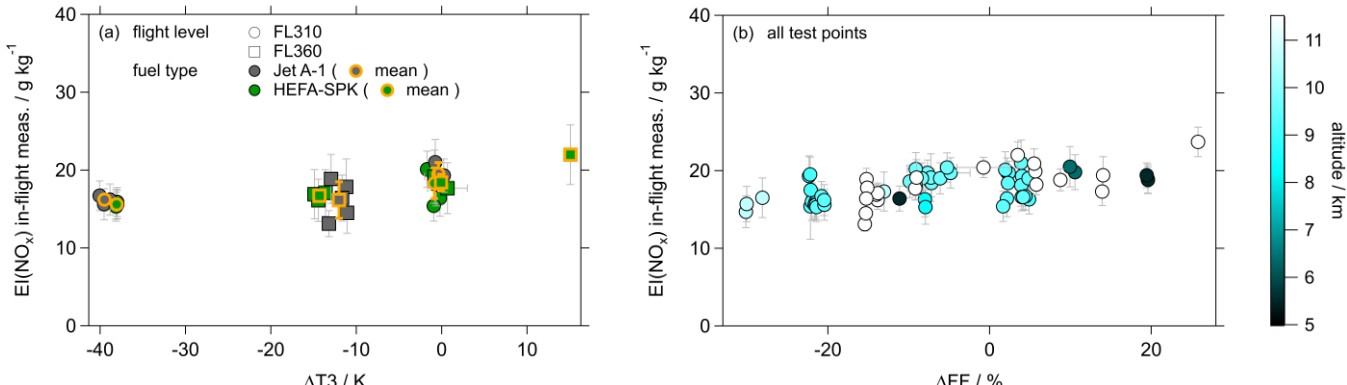

**Figure 5: EI(NOₓ) as calculated from near-field in-flight measurements behind the Airbus A350-941 with Rolls-Royce Trent XWB-84 engines (a) versus desensitized ΔT3 at constant conditions at FL310 and FL360, (b) versus desensitized ΔFF (fuel flow); color coded by altitude.**

Figure 5b presents data points acquired during the ECLIF3 experiment plotted against fuel flow range and color coded by flight altitude. Despite the scattered data, the expected increase in NOₓ emissions with increasing fuel flow can be recognized. Due to the multi-dimensional dependency of EI(NOₓ) on more than one engine parameter we cannot further assess the relative importance of individual engine parameter changes. However, we do not observe differences for the different flight altitudes where the measurements took place.

### 3.2 Comparison of in-flight ECLIF3 NOₓ emission indices with engine emission prediction methods

In this section, we compare our near-field ECLIF3 in-flight measurements with predictions. Please note that all ECLIF3-1 and -2 test points were analysed using BFFM2 and aptFFM2, whereas P3T3 method results are only available for ECLIF3-2. The uncertainty for BFFM2 is around 10% (DuBois and Paynter, 2006), for aptFFM2 ~20% (ICAO, 2020; Teoh et al., 2022), for

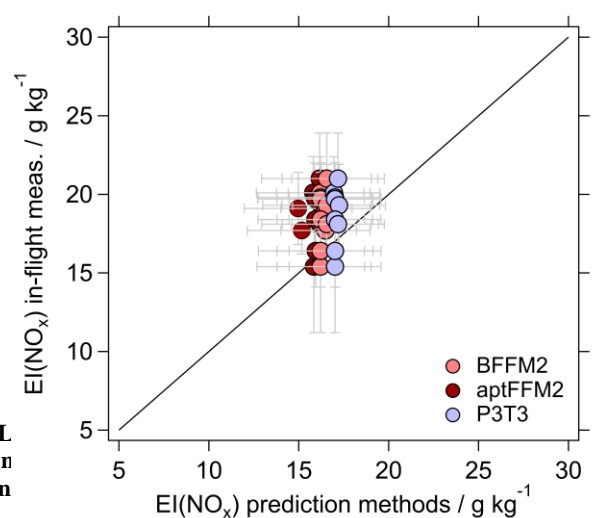

**Figure 6: EI(NOₓ) from ECL** ⋯ **the 0 K ΔT3 setting at FL310 and FL360.**
**Engine emission prediction** ⋯ **ft Engine Emissions Databank. The black**
**line marks the 1:1 agreemen** ⋯

P3T3 around 10 to 15% (SAE, 2009) and for the in-flight measured EI(NO$_x$) ~14% as derived from emission index uncertainty analysis below (see 3.4).

In general, the measurement to engine emission prediction method agreement is good with a correlation coefficient (R$^2$) of 0.3 to 0.4. For aptFFM2 / BFFM2 / P3T3 roughly 40 / 50 / 75 % of data points agree within a difference of ±3 g kg$^{-1}$ and are thus well within the combined errors of prediction method results and in-flight measurements. Deviations between the methods are within the error limits. However, predicted EI(NO$_x$) tend to be on average ~15% (aptFFM2, P3T3) to ~20% (BFFM2) lower than calculated EIs from the in-flight measurements in near field conditions. 6 shows, analogous to 5a, only data points for constant flight conditions at FL310 and FL360 with a focus on the ΔT3 setting of 0 K. This subset focuses on a set of conditions for which the DISA was quite similar, hence the atmospheric temperature conditions do not affect the emission indices. Still, the predictions tend to show smaller EI(NO$_x$) than determined by the in-flight measurements of the engines at cruise at slightly lower Mach numbers compared to typical cruise conditions.

### 3.3 Ground-based ECLIF3-2 NO$_x$ emission indices

The ground-based measurements behind the Rolls-Royce Trent XWB-84 engine were performed on two days at similar ambient temperatures. The reference measurements with fossil Jet A-1 were performed on 22 October 2021 (temperature: 12.2°C - 14.4°C; relative humidity: 87% - 81% during test run). The neat HEFA-SPK and an additional fuel blend (HEFA-SPK blended with a different Jet A-1) were tested on 23 October 2021 (temperature: 9.1°C - 15.2°C; relative humidity: 47% - 96% during test run). The hydrogen and carbon content by mass% of the blend are 14.39% and 85.61% respectively (see Table 1). Four test points with Jet A-1 were repeated on the second day in order to identify any biases from changes in environmental conditions or different probe alignment. The engine was operated at several different power settings ranging from idle conditions to maximum climb.

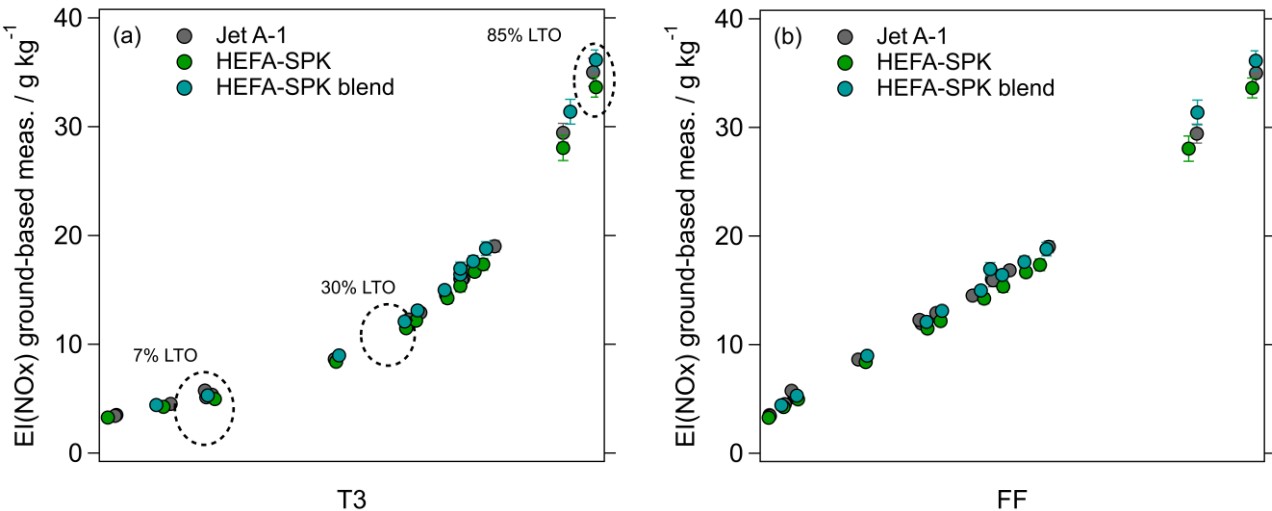

**Figure 7: EI(NOₓ) from ECLIF3-2 ground-based measurements behind the Airbus A350-941 with Rolls-Royce Trent XWB-84 engines versus T3 (referenced to 60% RH, (a)) and FF (fuel flow) (b) for Jet A-1, HEFA-SPK and Blend. Values on the x-axis are undisclosed. Black circles indicate LTO points.**

Thus, the ground-based measurements allow the detection of the EI(NOx) curve over a larger range of T3 and fuel flow (see Figure 7). The EI(NO$_x$) emission curve shows a continuous increase with higher thrust, and corresponding increasing T3, P3 and fuel flow. As expected and discussed above, NO$_x$ emissions are not significantly affected by the fuel composition. The measured EI(NO$_x$) for the different fuel types agree within estimated error margins. These findings are in line with results of ground-based measurements behind an Airbus A320-232 with IAE V2527-A5 engines in 2018 using fossil Jet A-1 fuels as

well as blends of HEFA-SPK and Jet A-1 as discussed by Schripp et al. (2022). They also found an independence of NO$_x$ emissions on fuel type, and similar sensitivities to combustion temperature ($\Delta$T3 ~40K, $\Delta$EI(NO$_x$) ~4 g kg$^{-1}$). Bulzan et al. (2010) also presented and discussed NO$_x$ emissions of a ground-based experiment in 2009 targeting the CFM56-2C1 engine of the NASA DC-8 aircraft burning pure fossil fuels (JP-8) as well as blends with Fischer-Tropsch fuels based on natural gas and coal. They found an increase of EI(NO$_x$) with fuel flow by about ~6 g kg$^{-1}$ per ~100% more burned fuel, which is in line

with the measurements performed in this study.

Figure 8 compares EI(NO$_x$) from far-field in-flight measurements with P3T3 and BFFM2 engine emission prediction methods that use the ECLIF3-2 ground-based measurements of the same Rolls-Royce Trent XWB-84 engine as input (see Figure 7). By using the engine emission prediction methods, the ECLIF3-2 ground-based measurements are related to in-flight conditions and cruise altitudes, i.e. to lower pressure and absolute humidity compared to the ground. Although lower absolute humidity

should increase NO$_x$ by 12%, in general, the prediction methods are expected to show lower NO$_x$ levels compared to the ground-based measurements due to the lower pressure at altitude. The predictions are compared to ECLIF3-2 far-field measurements, where the Airbus A350-941 and the Falcon were flying at typical cruise conditions and Mach numbers. The two engine emission prediction methods agree well with the measurements at cruise altitudes within the estimated

uncertainties. The BFFM2 predictions are typically 10% higher compared to P3T3 methods which has been found before for
this type of combustor, but are still within the ΔT3 ranges. The agreement between the prediction methods and the in-flight
measurements is significantly improved by using the ground-based EI(NO$_x$) measurements on the same engine instead of using
data from the ICAO Aircraft Engine Emissions Databank as it was done for the comparison in Figure 6. The use of ECLIF3-
2 ground-based measurement data on the same engine considers a potential slight change in engine performance of well-
maintained in-operation engines. The comparison to in-flight measurements at typical cruise conditions in terms of T3, P3,
AFR and Mach number also ensures a better comparability of the predictions well within the range tested in rig tests.

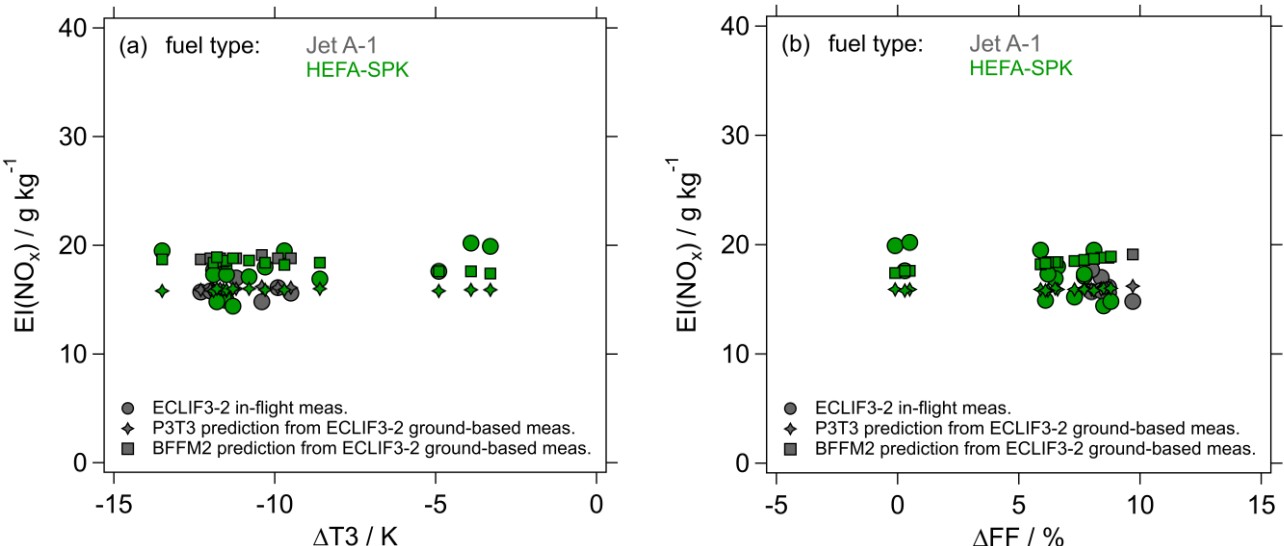

**Figure 8: EI(NO$_x$) (dots) from ECLIF3-2 in-flight measurements for different fuels at typical cruise Mach numbers in the far-field at FL320 and FL350 from behind the Airbus A350-941 with Rolls-Royce Trent XWB-84 engines versus desensitized ΔT3 (a) and desensitized ΔFF (fuel flow) (b). Predicted EI(NO$_x$) for the far-field measurement conditions from the P3T3 prediction method and**
**the BFFM2 prediction method calculated based on ECLIF3-2 ground-based measurements shown in Figure 7.**

### 3.4 In-flight emission index uncertainty analysis

Here, we present an in-depth analysis of different aspects contributing to the uncertainty of each individual inferred emission
index (see Eq. (1)) with respect to the in-flight measurements. The EI(NO$_x$) uncertainty consists of several individual errors:
(a) the uncertainty of the enhancement above an atmospheric background level ($\partial(\Delta NO_y)$ and $\partial(\Delta CO_2)$), which can be
subdivided into (a.a) the absolute accuracy of the measured species ($\partial NO_{y\_acc}$, $\partial CO_{2\_acc}$) and (a.b) the uncertainty related to the
atmospheric background determination ($\partial NO_{y\_bgr}$, $\partial CO_{2\_bgr}$); (b) the uncertainty in EI(CO$_2$); and (c) the uncertainty in the
molar masses of NO$_2$ and CO$_2$. The total uncertainty ($\partial EI(NOx)$) is then estimated using Gaussian error propagation following
Eq. (3):

$$\partial\text{EI}(\text{NO}x) = \pm \sqrt{\begin{array}{c}\left(\frac{\partial\text{EI}(\text{NO}x)}{\partial NOy}\partial NO_{y\_acc}\right)^2 + \left(\frac{\partial\text{EI}(\text{NO}x)}{\partial NOy}\partial NO_{y\_bgr}\right)^2 + \left(\frac{\partial\text{EI}(\text{NO}x)}{\partial CO_2}\partial CO_{2\_acc}\right)^2 + \left(\frac{\partial\text{EI}(\text{NO}x)}{\partial CO_2}\partial CO_{2\_bgr}\right)^2 + \\ + \left(\frac{\partial\text{EI}(\text{NO}x)}{\partial\text{EI}(CO_2)}\partial\text{EI}(CO_2)\right)^2 + \left(\frac{\partial\text{EI}(\text{NO}x)}{\partial M(NO_2)}\partial M(NO_2)\right)^2 + \left(\frac{\partial\text{EI}(\text{NO}x)}{\partial M(CO_2)}\partial M(CO_2)\right)^2\end{array}}$$

(3)

The individual uncertainty terms and the total uncertainty for EI(NO$_x$) are listed in Table 3. For EI(NO$_x$) the mean uncertainty from all in-flight plume encounters sums up to ~14%. Figure 9 further depicts the relative contribution of the individual terms to the total uncertainty for each plume encounter. It is evident that the most important uncertainty term for EI(NO$_x$) is the NO$_y$

accuracy. For future aircraft experiments we plan to implement a different dilution approach and suggest flying in a larger distance to the aircraft of interest to prevent the instrument from running into saturation effects.

**Table 3: Individual contributions to the total uncertainty $\partial\text{EI}(\text{NO}x)$ for all in-flight exhaust encounters. The table denotes mean values; however, the uncertainty is estimated for each individual plume encounter individually.**

| Uncertainty term | Mean uncertainty estimate / % |
|---|---|
| $\partial$NO$_y$ acc | 13 |
| $\partial$NO$_y$ bgr | <1 |
| $\partial$CO$_2$ acc | <1 |
| $\partial$CO$_2$ bgr | 1 |
| $\partial$EI(CO$_2$) | 0.1 |
| $\partial$M(NO$_2$) | 0.002 |
| $\partial$M(CO$_2$) | 0.003 |
| Total uncertainty $\partial$EI(NO$_x$) | 14 |

The measurement uncertainties, i.e. the measurement accuracies for NO$_y$ and CO$_2$ were described in Section 2.1, including errors from the individual instruments, measurement techniques and calibration procedures. These absolute measurement accuracies (in ppb and ppm) are then translated into a relative accuracy (in %) based on the maximum mixing ratio enhancement observed during each exhaust plume encounter. The mean relative accuracy for CO$_2$ is <1% (0.1-2.5%) and for

NO$_y$ 13% (9-23%), see Table 3. The contribution of the individual accuracy of NO$_y$ or CO$_2$ is generally higher (lower) when the encountered mixing ratio enhancement was low (high). The uncertainty of EI(CO$_2$), as well as of the molar masses, are negligible. The atmospheric background mole fraction needs to be determined for each individual plume encounter individually to account for horizontal, vertical as well as temporal gradients in the ambient atmosphere. However, the typical atmospheric background variation of NO$_y$ (4 ppb) and CO$_2$ (5 ppm) is not sensitive to the high mixing ratios encountered ($\Delta$NO$_y$ ~500-

4000 ppb, $\Delta$CO$_2$ ~100-800ppm). The atmospheric background itself is estimated based on the following assumption. Primarily,

prior to and after the plume encounter the probed air mass needs to be free of engine exhaust. Due to the natural dynamics of the troposphere, the atmospheric background for the long-lived gases $CO_2$ is not always as obvious as in the short-lived $NO_y$,

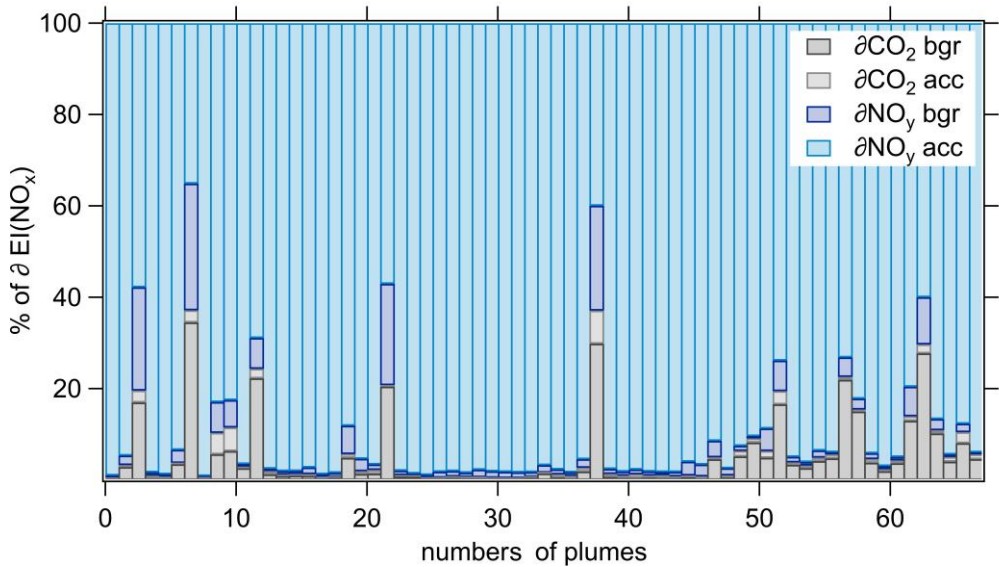

**Figure 9: Stacked relative individual uncertainties contributing to the total uncertainty $\partial EI(NO_x)$ for in-flight plume encounters.**

hence, $NO_y$ is taken as the main focus for this determination. If prior and after the encounter the mole fractions are identical, this value is taken for the atmospheric background. If the mole fractions are different, the atmospheric background is set in between these values. The mean difference from the mixing ratio at the start and end of the individual plume encounter to the respective atmospheric background estimate is considered in the uncertainty of the atmospheric background estimate. Moreover, the standard deviation ($1\sigma$) of an atmospheric background-like sequence is considered.

## 4. Conclusion

In conclusion, we presented the first $EI(NO_x)$ in-flight measurements for a modern long-range aircraft and engines since 1995 (Schulte et al., 1997). We showed that the measurements and methodology are adequate to infer $EI(NO_x)$ emissions from aircraft at high altitudes. As expected from previous ground engine tests, the fuel type, even a 100% HEFA-SPK, has no statistically significant effect on the $NO_x$ emission index. $EI(NO_x)$ increases with increasing combustion temperature, pressure, and fuel flow for the measured cruise T3 range conditions. Furthermore, the in-flight measurements generally agree with predictions from three different engine emission prediction methods within combined uncertainties when using data sets from the ground-based ICAO Aircraft Engine Emissions Databank as input, with a slight trend towards modelled lower emission indices. In order to consider performance variations of the operational engines during maintenance cycles and to avoid engine to engine performance variations we performed ground measurements behind the same engine over a wide T3 range. The

ground-based measurements show an increase in EI(NO$_x$) with increasing thrust, as explained by higher combustor temperatures. The ground-based measurements were then used as input to predict EI(NO$_x$) at cruise altitudes for typical cruise conditions using current engine emission prediction methods. The methods generally agree better with the measured EI(NO$_x$) for in-flight measurements at typical cruise conditions with respect to P3, T3, AFR and Mach number. These experiments present the first in-flight measurements targeting NO$_x$ emissions of latest-generation engines at high altitudes and thus provide a valuable data set of EI(NO$_x$) cruise measurements for the evaluation of state-of-the art engine emission prediction methods. The measurements thereby enhance the sparse existing data set of cruise emissions of older generation engines.

**Data Availability**

The CO$_2$ and NO$_y$ flight measurement data are released at https://doi.org/10.5281/zenodo.10646359 (Harlass et al., 2024).

**Author Contribution**

TH analyzed the NO$_y$ and CO$_2$ measurement data and wrote the manuscript with inputs from all co-authors. TH, RD, RM, DS, MS, SK and PSt performed the Falcon measurements. TS, TG and LB performed the ground-based measurements. CV and CR concepted and designed the aircraft experiments. US, MJ, DL, DA and PM performed the model simulations. All authors contributed to the interpretation of the data, reviewed the manuscript and have given approval to its final version.

**Competing Interests**

MJ, DL, PM and PS are employed by Rolls-Royce plc.; DA is employed by Rolls-Royce Deutschland; RS is employed by Neste Corporation. All other authors declare that they have no conflict of interest.

**Funding Sources**

TB was funded by the Deutsche Forschungsgemeinschaft (DFG, German Research Foundation) under project nr 510826369 and CV by DFG SPP 1294 HALO under project nr VO1504_9 and by European Union under Grant Agreement No 101101999 and No 101114613.

**Acknowledgments**

The authors especially thank DLR-FX for the experiment cooperation. This involves not only our pilots Michael Grossrubatscher, Thomas von Marwick, Philipp Weber and Roland Welser, but also the management Georg Dietz and Oliver Paxa and the group of Martin Zöger, Andreas Giez, Vladyslav Nenakhov, Christoph Grad, Marina Schimpf, Christian Mallaun, David Woudsma, Alexander Wolf, Frank Probst, Stefan Hempe and many more. We also thank the flight test team of Airbus for the great coordination of the aircraft on ground as well as in the test areas. Great recognition is dedicated to the Rolls-

Royce Technical Support team in Toulouse. And a special thanks is dedicated to all other ECLIF3 partners for their helpful cooperation. We would also like to thank Roger Teoh and Mark Stettler, who provided the source code for modelling emission indices. The first author really appreciates the great effort of Tiziana Bräuer and Magdalena Pühl, who finished the manuscript and answered the reviewer comments during my parental leave.

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
