# Peer review of "Measurement report: In-flight and ground-based measurements of nitrogen oxide emissions from latest generation jet engines and 100% sustainable aviation fuel"

_EGUsphere, 2024_

## Author Response (AR1)

**Author's Response to Referees**

**Measurement report: In-flight and ground-based measurements of nitrogen oxide emissions from latest generation jet engines and 100% sustainable aviation fuel**

**RC1**: 'Comment on egusphere-2024-454', Anonymous Referee #1, 03 Apr 2024

We would like to thank the Referee for reviewing our manuscript. We improved the figures and Table 1 to make them more understandable and intuitive and also introduced consistent terminology in the text. We separated the method description for in-flight and ground-based measurements in Section 2 (Material and Methods). The answers to specific comments are addressed below (for convenience the Referee comments are repeated in italic letters).

**Comment 1):** *A clarification is needed what is measured and reported in the specific sections and plots. From my understanding for all altitude measurements NOy is measured and reported for all ground level measurements NOx is reported. Since NOx ≠ NOy even at the engine exit (HONO can make up 6% fraction e.g. dx.doi.org/10.1021/es200921t) the actual reported data needs to be better described (or not labeled as NOx) throughout the manuscript. Maybe I misunderstood, but also then it needs a clarification.*

**Answer comment 1):** We can confirm that in-flight total reactive nitrogen NOy is measured by a chemiluminescence detector during ECLIF3 and not only NOx. The same is the case for the ground-based measurements. The ground-based measurements are now clearer explained in the new section 2.3 on page 6. For both in-flight and ground-based measurements, the emission indices for NOx are calculated by using the NOy concentration and the molar mass of $NO_2$, assuming that only a small fraction of NOx is reacting to other reactive nitrogen oxides. Concerning HONO, we added Lee et al. (2011) as a reference. Thank you for suggesting this paper. Because Lee et al. (2011) refers only to ground-based measurements and an older test engine, we also added Jurkat et al. (2011) as a reference on page 7, line 78 of the paper. Jurkat et al. (2011) reports in-flight measurements of HONO/NOy behind 8 different aircraft to be up to 3.6%. We are aware that by using NOy instead of NOx, EI(NOx) might be slightly overestimated, but fractions of HONO or $HNO_3$ are assumed to be below the total uncertainty for in-flight EI(NOx) of 14%, that we estimated in this paper. Concerning the labeling, we made sure that we clearly separate between NOy measurements and EI(NOx).

**Comment 2) Table 1:** *Please add energy content, aromatic content, and density of fuels. The information on the used fuel is rather sparse – while they energy content might not vary much probably around 42- 43 MJ/kg it might provide a better insight why not much difference was observed between the fuels? The other parameters are needed to link the information provided to other studies and the ICAO database – this is really needed for putting the results into context, the study has otherwise only a limited value.*

**Answer comment 2):** We added information on density and aromatics content to Table 1 and to the corresponding text as proposed. We did not add a column on the energy content of the fuels, as it does not differ significantly. This is described now in the text. We also added a column describing the numbers of test points to the table.

**Comment 3) Figure 3:** *are the blue points from EDB or your measurements? Please clarify.*

**Answer comment 3):** The blue points are from the ICAO EEDB. We improved the legend and caption of Figure 3 to make it better understandable.

**Comment 4) Figure 6**: *How do these data compare to the ICAO EDB? Are any of the ICAO Annex16 corrections applied to the data or is this raw? Please clarify.*

**Answer comment 4):** Unfortunately, the ground-based ECLIF3 measurements cannot be compared in Figure 6 with the ICAO EEDB, as T3 and the fuel flow are undisclosed for the ground-based ECLIF3 measurements and a comparison would make the actual values comprehensible. No ICAO Annex16 corrections (temperature compensation or sea-level correction) are applied to the ground-based measurements, but the setup is based on ICAO.

**RC2**: 'Comment on egusphere-2024-454', Anonymous Referee #2, 22 Apr 2024

We like to thank the Referee for reviewing our manuscript. We agree that the paper is complex as we are combining so many different methods. We separated the method description of in-flight and ground-based measurements in Section 2 (Material and Methods) and improved the legends, captions and axis titles of our figures to make them better understandable. Also, we introduced consistent terminology in the text and adapted the section headlines. The answers to specific comments are addressed below (for convenience the Referee comments are repeated in italic letters).

**Comment 1):** *The paper is at times hard to follow. The authors could clarify a bit better throughout what exactly is being shown or discussed. For example, in Figure 5, are the model results obtained using data from the ground tests?*

**Answer comment 1):** We focused on improving the clarity of the paper and improved the caption and axis titles of Figure 5 to make clear that the prediction methods use the EI(NOx) from the ground-based ICAO Aircraft Engine Emissions Databank. We also improved legends, captions and axis titles of Figure 3, 4, 6 and 7 to make them more intuitive.

**Comment 2):** *The fact that the paper contains both in-flight and ground measurements is in my opinion one of its strongest points, and the authors could further build on that. For example, adding an equivalent to Figure 1 for the ground tests would be helpful, and expanding the description of what exactly was measured on the ground, how, at what distance, and how many times.*

**Answer comment 2):** We understand that clear information on the ground-based ECLIF3-2 measurements are needed. So, we brought all information on the method of the ground-based measurements together in a new section 2.3 *Ground-based NOy and CO2 measurement methods during ECLIF3*-2. It mainly consists of text that was previously elsewhere in the paper. For example, at what distance and how many times the ground-based test points were measured was previously described in the results section. But we also added information and a new figure at page 7. However, the main focus of the paper should remain on the in-flight measurements.

**Comment 3) Section 3.4:** *why is this analysis not performed for the ground measurements (or if it is please clarify)?*

**Answer comment 3):** Section 3.4 gives an uncertainty analysis for the in-flight emission indices due to the strongly varying ambient conditions during an in-flight measurement. A comparable analysis was not done for the ground-based measurements as the conditions are much more stable on ground.

**Comment 4):** *Putting the results in the context of ICAO's Aircraft Engine Emissions Databank would make this work more impactful (e.g. a comparison or equivalent that preserves any sensitive data).*

**Answer comment 4):** In the paper, the EI(NOx) for the measured engine from the ICAO Aircraft Engine Emissions Databank are depicted in Figure 3. A direct comparison with ground-based ECLIF3-2 measurements would made T3 and the fuel flow comprehensible and this data is sensitive. Beyond that, the in-flight measurement results are put indirectly in the context of ICAO's Aircraft Engine Emissions Databank in Figure 5 by comparing EI(NOx) from in-flight measurements with prediction methods that are based on the ICAO Aircraft Engine Emissions Databank.

**Minor comment a):** *Some typos: line 253, Table 2 T2 unit, x-axis of Figure 6, x-axis of Figure 8*

**Answer minor comment a):** Typos were corrected. Thank you for the comment.

**Minor comment b):** *It would be helpful to add to Table 1 the number of observations for each case (i.e. how many of the "points" that appear in following plots, e.g. Figure 4, are from each), separating the ground from the in-flight measurements for the ECLIF3-2 case.*

**Answer minor comment b):** We added a column with the numbers of test points for each fuel type to Table 1. For in-flight measurements test points are defined as plume encounters, for ground-based measurements test points are defined as an averaged measurement sequence at stable T3 operating conditions.

**Minor comment c):** *The data repo would benefit from a README or equivalent file. Despite the individual files having headers, an overview of the data provided would be helpful. For example, does it only include the in-flight measurements?*

**Answer minor comment c):** We added a README to the data repo.

---

## Author Response (AR2)

**Author's Response to Referees**

**Measurement report: In-flight and ground-based measurements of nitrogen oxide emissions from latest generation jet engines and 100% sustainable aviation fuel**

**Report #1, Submitted on 18 Aug 2024**

**Anonymous referee #3:**

The measurements reported in this manuscript are important. The authors have reasonably addressed most of the comments/concerns raised except one on NOy vs NOx by reviewer #1.

RC1 comment 1: "A clarification is needed what is measured and reported in the specific sections and plots. From my understanding for all altitude measurements NOy is measured and reported for all ground level measurements NOx is reported. Since NOx ≠ NOy even at the engine exit (HONO can make up 6% fraction e.g. dx.doi.org/10.1021/es200921t) the actual reported data needs to be better described (or not labelled as NOx) throughout the manuscript. Maybe I misunderstood, but also then it needs a clarification."

In the response, the authors confirmed that NOy was measured for both in-flight and ground level measurements. They explained that "the emission indices for NOx are calculated by using the NOy concentration and the molar mass of NO2, assuming that only a small fraction of NOx is reacting to other reactive nitrogen oxides." The HONO/NOy fraction can be up to 6% based on Lee et al. (2011) and up to 3.6% according to Jurkat et al. (2011). The authors' calculation basically assumes NOx=NOy (Equ. 1), and they justify this by arguing that "fractions of HONO or HNO3 are assumed to be below the total uncertainty for in-flight EI(NOx) of 14%." The authors acknowledged that "by using NOy instead of NOx, EI(NOx) might be slightly overestimated."

It is unclear to me what is the NOx/NOy ratio for the engine studied in this study and how the ratio may vary with power setting (and fuel types). Since NOy was measured and this is a measurement report manuscript, I think that it is more appropriate to report the measured values as NOy. The authors can point out in the abstract and main text that NOx is expected to be close to NOy and the uncertainty/underestimation induced by treating NOy as NOx is likely small and within the measurement uncertainty.

**Answer to Comment:**

First of all, we like to thank the referee for his comment. Unfortunately, the NOx/NOy ratio cannot be determined with the methods used during ECLIF3 and we cannot determine how the ratio may vary with power setting or fuel types. In the paper, we always use NOy when relating to the measured species and EI(NOX) when we talk about the emission index as we specifically calculated the emission index from the NO2 molar mass according to definition. We hope the wording is very precise enough in this way.

To make it clearer and more understandable what is measured and how EI(NOx) is calculated, we added following sentence to section 2.1 In-flight NOy and CO2 measurement methods during ECLIF3, line 84: "The instrument offers no measurement of NOx or the NOx/NOy ratio."

And we added following explanation to section 2.4 Emission index calculation and plume definition, line 189: "During ECLIF3, only NOy and no NOx concentrations were measured aboard the Falcon. NOx concentrations are expected to be close to NOy and the fraction of nitrogen acids in the exhaust gas is

assumed to be smaller than the NOy mean measurement accuracy. Hence, all reactive nitrogen species in the exhaust are detected and related to the initial NOx emissions."